# Developing a Holistic Success Model for Sustainable E-Learning: A Structural Equation Modeling Approach



**Ahmad Samed Al-Adwan** [1] **, Nour Awni Albelbisi** [2] **, Omar Hujran** [3,*] **, Waleed Mugahed Al-Rahmi** [4] **and Ali Alkhalifah** [5]

1 Electronic Business and Commerce Department, Business School, Al-Ahliyya Amman University, Amman 19328, Jordan; a.adwan@ammanu.edu.jo
2 Faculty of Education, University of Malaya, Kuala Lumpur 50603, Malaysia; noorbelbisi@gmail.com
3 Department of Analytics in the Digital Era, College of Business and Economics, United Arab Emirates University, Al Ain 15551, United Arab Emirates
4 Self-Development Skills Department, College of Common First Year, King Saud University, Riyadh 11451, Saudi Arabia; waleed.alrahmi@yahoo.com
5 Information Technology Department, College of Computer, Qassim University, Buraydah 52571, Saudi Arabia; a.alkhalifah@qu.edu.sa
* Correspondence: o.hujran@uaeu.ac.ae

**Abstract:** In higher education learning, e-learning systems have become renowned tools worldwide. The evident importance of e-learning in higher education has resulted in a prenominal increase in the number of e-learning systems delivering various forms of services, especially when traditional education (face-to-face) was suddenly forced to move online due to the COVID-19 outbreak. Accordingly, assessing e-learning systems is pivotal in the interest of effective use and successful implementation. By relying on the related literature review, an extensive model is developed by integrating the information system success model (ISSM) and the technology acceptance model (TAM) to illustrate key factors that influence the success of e-learning systems. Based on the proposed model, theory-based hypotheses are tested through structural equation modeling employing empirical data gathered through a survey questionnaire of 537 students from three private universities in Jordan. The findings demonstrate that quality factors, including instructor, technical system, support service, educational systems, and course content quality, have a direct positive influence on students' satisfaction, perceived usefulness, and system use. Moreover, self-regulated learning negatively affects students' satisfaction, perceived usefulness, and system use. Students' satisfaction, perceived usefulness, and system use are key predictors of their academic performance. These findings provide e-learning stakeholders with important implications that guarantee the effective, successful use of e-learning that positively affects students' learning.

**Keywords:** e-learning; success factors; information quality; COVID-19; TAM; system quality; ISSM; IS success model; e-learning adoption

## 1. Introduction

Most economic aspects have been affected by the development of information technology [1], which has contributed to tremendous change in the educational sector. Students' increased use of the web and Internet has encouraged educational institutions to replace traditional learning and teaching approaches. Consequently, higher education is now changing, and higher education institutions (HEIs) (i.e., universities) must accordingly fulfil students' requirements, needs, and expectations [2]. E-learning is essential to enable and execute these activities as it allows students to access learning resources regardless of location and time. According to Cidral et al. [3], e-learning systems offer personified, flexible learning; reduce the cost of learning; and enable learning on demand. Al-Fraihat [4] (p. 57) states that e-learning has "a significant role in shifting from teacher-centered to student-centered education."

Although many e-learning initiatives have been successfully implemented, most of them did not attain their intended objectives, progressed slowly, and observed increased dropout rates [4]. Additionally, assessing the success of e-learning represents a concern for e-learning systems' stakeholders. HEIs have substantially invested in e-learning systems to enhance and support learning [5]. However, integrating innovative e-learning systems to support both teaching and learning is a major challenge for HEIs. An argument has been that the value of investing in e-learning systems significantly relies on the implementation of these systems by instructors and students [6]. Students' acceptance and use of e-learning systems determine their success, and poor use of such systems prevents realizing their benefits, which leads to unsuccessful e-learning systems and thus poor return on investment [7]. These assumptions hold true especially in light of the rapid, erratic, and sudden outbreak of COVID-19. While the role of technology in supporting people in ordinary conditions is widely recognized, it is unclear the extent to which technology is capable to respond to their needs in unusual circumstances and extreme conditions such as COVID-19 [8].

Consequently, to help e-learning stakeholders successfully implement e-learning systems, many researchers have explored the key success factors of e-learning. Although these researchers have attempted to identify the requirements of e-learning systems, they have not revealed all the requirements [4]. The most significant factors that measure the success of e-learning systems are diverging in literature. Few studies have considered developing a holistic model that evaluates the success of e-learning systems from various angles. As Eom and Ashill [9] suggest, the holistic success model of e-learning should cover multiple levels of success. E-learning systems are a form of information system (IS) that incorporates human factors (i.e., learners and instructors) and nonhuman factors (i.e., learning management systems). Thus, examining various aspects of success related to human and nonhuman factors is vital. Al-Fraihat et al. [10] and Cidral et al. [3] have identified two prominent streams of e-learning adoption-related research. One stream focuses on adoption behaviors (e.g., system use, adoption, intention to use, usability), technological aspects, and the system. Nevertheless, because technology has become widely accessible and reliable, a more recent research stream has explored instructors' and students' interactions and attitudes as critical determinants of e-learning success. Accordingly, this study develops a holistic success model of e-learning that considers human and technological perspectives of e-learning.

## 2. Significance of the Study

The literature on e-learning is focused on e-learning adoption and postadoption. Two streams of research are related to the adoption and postadoption of e-learning. The first stream of research assumes users' postadoption behavior of e-learning systems as an extension of the initial acceptance behavior of e-learning systems and employs the same variables to explain acceptance and continued use [1,11–16]. Studies related to this stream have mainly used the technology acceptance model (TAM) [17] as the underlying theoretical framework and extended it using other supplementary theories and models, such as the unified theory of acceptance and use of technology (UTAUT) [18], UTAUT2 [19], and the theory of planned behavior (TPB) [20], to examine the adoption and continuance behavior of e-learning systems. The second stream of research has used the expectation confirmation model (ECM) of Bhattacherjee [21] as the focal theoretical base [22–24]. Researchers of the second stream have been inclined to integrate various other frameworks, such as the IS success model (ISSM) [25], TPB [20], and TAM [17] along with the IS continuance model.

However, the aforementioned streams' studies have primarily assessed factors that predict e-learning systems' adoption and postadoption and do not consider how such factors or the actual use of e-learning systems relates to learning outcomes. To fill this gap in the literature, many scholars have extended their investigation beyond adoption behaviors (i.e., use or continued use) and have examined the impact of adoption behaviors on e-learning use outcomes. In this regard, two schools of research have emerged. In the first school, self-developed models and frameworks have been employed as the theoretical

base [26–28]. Such studies have been conducted with different outcome variables that use various explanatory variables, resulting in models with a weak connection to theory. Accordingly, this weakness makes generalizing the findings of these studies difficult. Consequently, these studies are insufficient to explain how e-learning use and its determinants affect the outcomes of e-learning use. Conversely, scholars of the second school have relied on the ISSM, which offers theoretical support for the association between the behaviors of e-learning adoption and outcomes. However, an argument is that the ISSM has limited theoretical support concerning the relationships between determinants and behaviors of e-learning adoption [29].

Therefore, this study first indicates the limitations of research on technology adoption and IS success and then applies a combination of these research directions to overcome their limitations. A reasonable endeavor is to pursue theoretical support from both research streams to develop a holistic model that considers the determinants, behaviors, and outcomes of e-learning adoption. Specifically, the proposed model is theoretically grounded in TAM and is called the ISSM. Thus, the proposed model retains the quality factors of the ISSM as the main determinants of e-learning adoption behavior (i.e., use). Additionally, the model borrows two constructs from TAM—perceived usefulness (PU) and perceived ease of use (PEOU)—as further key determinants of e-learning use behavior. The conceptualization of the proposed model is discussed in detail in Section 4.

### 3. Information System Success Model (ISSM)

DeLone and McLean [30] developed the model of IS success. This model comprises six success factors that measure IS success, namely, organizational impact, individual impact, use, satisfaction (SAT), information quality, and system quality. The model suggests that quality factors (system and information quality) directly influence user satisfaction and IS use. Additionally, satisfaction and system use affect the individual impact and subsequently influence organizational impact. However, DeLone and McLean [25] updated and modified the original model by adding another quality factor—service quality—to predict IS use and user SAT (see Figure 1). Another modification integrates organizational and individual impacts into one factor called net benefits. The updated model of IS success (D&M model) has captured scholars' attention in the field of IS. This demonstrates that IS success can be assessed using a set of quality factors (i.e., service quality, systems quality, and information quality). Subsequently, these factors influence user satisfaction and use. The D&M model theoretically indicates that a high-quality IS results in high user satisfaction and thus high levels of IS use and an increased perception of net benefits.

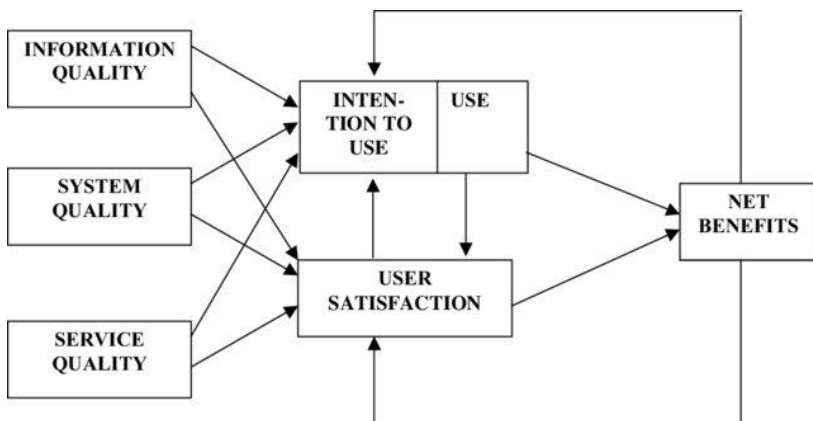

**Figure 1.** Information system success model (DeLone and McLean [25]).

Although the D&M model has been investigated in various areas of IS, the model has limitations, particularly in e-learning-related research. Many studies [31,32] have confirmed that the determining factors of the net benefits variable (the outcome construct of the D&M model) are not abundantly acknowledged. This assertion holds true, especially because

the net benefits variable is case-specific and therefore differs based on the requirements of each user and entirely relies on the objectives and type of a particular information system. Consequently, additional research is necessary to identify additional determining factors of net benefits, particularly in the educational technology context. Thus, this study suggests that self-regulated learning (SRL) and PU are additional determinants of the net benefits variable.

Various studies that employed the D&M model in the e-learning context have reported diverse estimates of variance explained ($R^2$) by quality factors. Eom et al. [33] (p. 158) observes that "the DeLone and McLean model has limited explanatory power for explaining the role of e-learning systems on the outcomes of e-learning." Hence, to enhance the explanatory power of the D&M model, scholars have indicated that additional research is required to explore additional quality factors of e-learning systems [10]. This study responds to these requests for further research by contextualizing and extending the original constructs of the D&M model to fit the e-learning settings. The contextualization and the operationalization of the proposed model are discussed in the next section.

### 4. Theoretical Foundation

This study suggests a contextualized model of e-learning success factors (see Figure 2) that is based on the D&M model and TAM. The proposed model is expected to be a driver of the development, design, and delivery of e-learning systems initiatives. Recognizing the aforementioned applications of the D&M model from the literature, this study adopts and contextualizes the key relationships of the D&M model's constructs into the setting of this study, that is, e-learning. Specifically, because of the aforementioned limitations, the following modifications have been made.

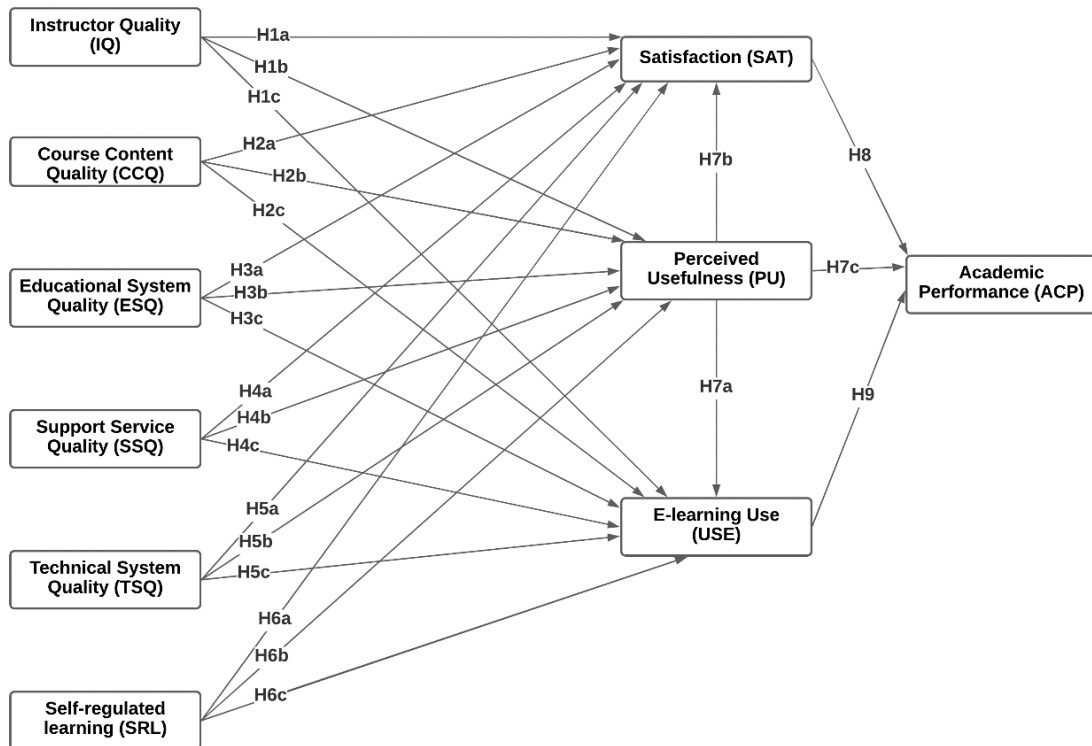

**Figure 2.** The research model.

- According to Cheng [34], in the e-learning environment, service quality is the assistance provided by instructors and support service technicians. Accordingly, service quality in e-learning can be decomposed into two key dimensions: instructor quality (IQ) and support service quality (SSQ).

- System quality is a vital indicator of e-learning quality and has two dimensions: educational system quality (ESQ) and technical system quality (TSQ). ESQ is related to the presence of education-related features such as diverse learning styles, evaluation styles, and communication and interactivity tools [35]. TSQ is concerned with technology-related aspects such as usability, availability, and reliability. Davis [17] demonstrates that PEOU reflects the extent to which users perceive a system's use as effortless. Hence, PEOU is a key indicator of TSQ.
- Similar to its use in the literature [36,37], information quality is used in the D&M model as a general quality measure of IS success. Thus, to produce a more fitting model that adapts to the requirements of e-learning systems, information quality is reworded as course content quality (CCQ).
- The proposed model includes SRL as another critical factor in e-learning environments. SRL is a major success factor in e-learning environments [38]. It represents a context-specific dimension that reflects students' ability to achieve their learning objectives through controlling their motivations and learning behaviors [39].
- The proposed model retains the original relationships of the D&M model, except the relationship between user SAT and USE, which has been criticized and regarded as confusing and theoretically weak [10,40]. Therefore, in the proposed model, PU mediates the relationship between USE and user SAT.
- This study facilitates the building of a context-specific model that adheres to the requirements and specifications of e-learning success by replacing the construct of net benefits with that of academic performance (ACP) as the outcome variable.

In summary, the quality factors and SRL described act as external variables that affect PU, user SAT, and USE. Additionally, PU directly influences user SAT, ACP, and USE. ACP is influenced by PU, user SAT, and USE. The following section explains the proposed hypotheses of the research model.

## 5. Hypothesis Development

### 5.1. Instructor Quality

Cheng [34] (p. 237) defines IQ as "the degree to which learners perceive that the instructor's attitude that relates to the instructor's response timeliness, teaching style, and help toward learners via the e-learning system." IQ represents instructors' teaching styles and attitudes that discernibly affect learners' participation, attitudes, and enthusiasm toward e-learning systems. Pham et al. [41] state that the instructors' feedback is a crucial factor in online classes; moreover, learners' perception of quality and timeliness of instructor feedback is pivotal in the success of online courses. Additionally, Rajabalee and Santally [42] observe that instructor support is a crucial element in shaping learners' SAT. Specifically, learners in e-learning environments may sense frustration and express negative feelings when they receive inadequate instructor support even if they perform well.

Additionally, appropriate instructor involvement in active academic guidance has been reported to directly contribute to developing learners' SAT, performance, and achievement. Addressing learners' needs or problems in a quick, efficient manner; providing online feedback on learners' activities in e-learning systems (i.e., assignments); and sharing significant information on the online course are indispensable aspects to ensure learners' SAT with e-learning systems [43]. Lee et al. [44] emphasize that providing proper academic guidance increases learners' interest in high achievements and desire toward self-improvement. Accordingly, when instructors are involved in active academic guidance and provide quality and timely feedback using e-learning systems, learners' acceptance and motivation to use such systems increases. Instructors in e-learning environments are responsible for setting learning goals and providing learning materials and activities (i.e., quizzes, assignments) to accomplish those goals. Accordingly, instructors' technical competencies have been suggested to be critical in integrating ICT into the educational process [45,46]. This finding indicates that instructors must possess proper pedagogical and technological knowledge. Such competencies enable instructors to engage and be

involved effectively in active academic guidance, which increases learners' usefulness perception of e-learning systems [43]. As a result, such a high IQ will increase learners' SAT and their usefulness perception of e-learning systems.

Thus, Hypothesis 1 (H1) is proposed:

**Hypothesis 1a (H1a).** *IQ positively influences perceived SAT with the e-learning system.*

**Hypothesis 1b (H1b).** *IQ positively influences PU of the e-learning system.*

**Hypothesis 1c (H1c).** *IQ positively influences the use of the e-learning system.*

### 5.2. Course Content Quality

Information quality assesses the quality of the information provided by IS [25]. Information quality of an e-learning system is a fundamental aspect in assessing the success of the system as poor information quality may generate serious problems in attaining learning goals [10]. Information quality in the context of e-learning represents CCQ. Notably, CCQ is the content quality provided by an e-learning system [37,47]. The key attributes of CCQ are accuracy, usefulness, reliability, comprehensibility, availability, relevancy, completeness, and being up-to-date. Updated, comprehensive content fulfils learners' expectations, making them feel pleased with the e-learning systems [34]. Additionally, e-learning systems are perceived as useful when constantly updated, rich course content is provided, and the course content can be customized to learners' needs. Mtebe and Raisamo [48] explain that well-conceived courses that address intended learning outcomes enable learners to perform effectively in courses delivered by e-learning systems. Hence, a suggestion is that e-learning systems with high-quality course content would support learners in improving their outcomes and grades. Additionally, these courses with high-quality content would encourage learners to persist in using such systems, thereby increasing their SAT levels. Similarly, Yakubu and Dasuki [1] assert that CCQ is pivotal in motivating learners to use e-learning systems by improving their SAT with such systems.

Thus, Hypothesis 2 (H2) is proposed:

**Hypothesis 2a (H2a).** *CCQ positively influences perceived SAT with the e-learning system.*

**Hypothesis 2b (H2b).** *CCQ positively influences PU of the e-learning system.*

**Hypothesis 2c (H2c).** *CCQ positively influences the use of the e-learning system.*

### 5.3. Educational System Quality

ESQ is a necessary element for achieving the targeted goals set by institutions [36]. It denotes the degree to which the features of e-learning are desirable in educational settings. ESQ focuses on measuring the quality of e-learning systems in terms of the presence of features such as communication facilities (i.e., chatting), learning styles' diversity, evaluation material, and collaboration and interactivity methods [7,10]. The literature has demonstrated that educational tools and features of e-learning systems such as active learning; adequate information sharing; efficient, effective communication and collaborative tools (i.e., chatrooms, discussion forums); storage; and document sharing can remarkably maximize e-learning systems use and learners' SAT [3,36,37]. Moreover, Goh et al. [49] reveal that interaction with peers and instructors through e-learning systems is a key aspect that determines SAT and facilitates accomplishing learning outcomes. Additionally, there is a significant connection between ESQ and PU [50].

Thus, Hypothesis 3 (H3) is proposed:

**Hypothesis 3a (H3a).** *ESQ positively influences students' SAT with the e-learning system.*

**Hypothesis 3b (H3b).** *ESQ positively influences PU of the e-learning system.*

**Hypothesis 3c (H3c).** *ESQ positively influences the e-learning USE.*

*5.4. Support Service Quality*

　　Cheng [34] (p. 237) defines SSQ as the "degree to which a learner perceives that the overall quality of personal support services from the e-learning system." SSQ reflects the quality of assistance and support delivered to users by IT technicians and the IS unit (i.e., helpdesk, training). Lee [51] and Pham et al. [41] state that SSQ is a key determinant of learners' SAT and acceptance of e-learning systems. The literature has demonstrated that offering suitable support services by services administrators or a helpdesk for e-learning systems increases learners' perception of the usefulness, SAT, and acceptance of e-learning systems [43,52,53]. Cheok and Wong [54] assert that technical support is fundamental in using e-learning systems. Insufficient technical support may cause frustration and problems for learners during their interactions with e-learning systems, leading learners to perceive that the benefits of such systems do not outweigh the problems [24]. Technical support is an important factor that facilitates integrating and using technology in learning. Turugare and Rudhumbu [45] suggest that the availability of technical support to assist in adopting e-learning systems is a critical success factor that ensures that instructors and learners will not have to manage technical difficulties beyond their capabilities. Thus, technical support plays a critical role in conceiving the usefulness of the various functions of e-learning systems and thus enhances learners' performance and interaction with systems. Literature has anticipated that offering services by IT technicians and units in organizations related to an e-learning system considerably affects learners' perception of the system's usefulness; subsequently, usefulness perception influences learners' perceived SAT with the system [10]. Technical support participates effectively in generating favorable perceptions toward e-learning systems as it makes learners believe that the benefits of using e-learning systems are worth the effort.

　　Thus, Hypothesis 4 (H4) is proposed:

**Hypothesis 4a (H4a).** *SSQ positively influences perceived SAT with the e-learning system.*

**Hypothesis 4b (H4b).** *SSQ positively influences PU of the e-learning system.*

**Hypothesis 4c (H4c).** *SSQ positively influences the use of the e-learning system.*

*5.5. Technical System Quality*

　　According to Seta et al. [37] and Lee and Jeon [47], system quality denotes the desirable functionalities and technical characteristics of IS. It relates to the efficiency, accuracy, and technical success of IS because it accounts for the absence or existence of bugs in the systems [35]. TSQ of an e-learning system is evaluated by measuring the quality of technical characteristics such as security; ease of navigation; availability; user friendliness; reliability; ability to integrate with other systems; ease of access; ease of finding information; and efficiency of functionalities; and it prompts a structured, interactive design [1,37,55]. Additionally, a suggestion is that TSQ is assessed by the extent to which the functionalities of an e-learning system are easy to learn [4]. The presence of friendly user interfaces and modern graphical interfaces improve learners' SAT and stimulate their interest in using the system [56]. The high technical quality of an e-learning system positively influences learners' perceptions that the system can deliver useful, reliable functionalities for learning activities, which increases learners' SAT, and they thus become keen on the system. System quality is a key determent of learners' SAT [10], USE [1], and usefulness [55].

　　Thus, Hypothesis 5 (H5) is proposed:

**Hypothesis 5a (H5a).** *TSQ positively influences perceived SAT with the e-learning system.*

**Hypothesis 5b (H5b).** *TSQ positively influences PU of the e-learning system.*

**Hypothesis 5c (H5c).** *TSQ positively influences the USE of the e-learning system.*

*5.6. Self-Regulated Learning*

Zimmerman [57] (p. 541) describes SRL as "metacognitive, motivational, and behavioral processes that are personally initiated to acquire knowledge and skill, such as goal setting, planning, learning strategies, self-reinforcement, self-recording, and self-instruction." Similarly, Landrum [58] states that self-regulation is attained when students can employ self-managing actions and independently implement learning processes. Al-Adwan et al. [38] indicate that SRL promotes autonomous, self-directed, and independent learning. Self-regulated students are dedicated participants who competently control the activities of their learning. For instance, self-regulated students monitor their learning processes, organize and review the content to be learned, manage their time, set up and follow plans for accomplishing all required learning-related tasks on time, and maintain positive motivational beliefs in their abilities and the value of learning [59]. SRL is a necessary success factor for e-learning, namely, students must possess a high level of autonomy in their learning, especially in the low presence of instructors or peers [60]. Different from the traditional (face-to-face) learning mode, in the new mode, the low level of presence of instructors shifts the task of learning control from instructors to individual students. Tasks executed by instructors (i.e., setting learning goals and progress assessment) are becoming the duty of the individual student. Accordingly, students with low SRL level would experience major difficulties in such considerably autonomous learning settings; thus, they would become dissatisfied, view the e-learning system as not useful, and resist using it.

Thus, Hypothesis 6 (H6) is proposed:

**Hypothesis 6a (H6a).** *SRL positively influences students' SAT with the e-learning system.*

**Hypothesis 6b (H6b).** *SRL positively influences students' PU of the e-learning system.*

**Hypothesis 6c (H6c).** *SRL positively influences the USE of the e-learning system.*

*5.7. Perceived Usefulness*

Davis [17] (p. 320) defines PU as "the degree to which a person believes that using a particular system would enhance his/her job performance." In TAM-related research, PU has been remarkably salient as a primary determinant of technology acceptance. It reflects the instrumental value of IS, such as e-learning systems. E-learning systems are expected to deliver useful features such as downloading learning material and interacting with peers and instructor that improve students' learning. Literature has indicated that PU positively influences students' SAT, USE, and ACP [10,61–63]. When students perceive that the e-learning system benefits their learning, they are inclined to be satisfied and subsequently inclined to use it. Moreover, when the system adds value by facilitating students' in achieving higher performance and attaining learning goals, their ACP is enhanced.

Thus, Hypothesis 7 (H7) is proposed:

**Hypothesis 7a (H7a).** *PU positively influences e-learning USE.*

**Hypothesis 7b (H7b).** *PU positively influences students' SAT with the e-learning system.*

**Hypothesis 7c (H7c).** *PU positively influences students' ACP.*

*5.8. Satisfaction*

SAT is a vital factor in assessing the success of IS [64,65]. Salam and Farooq [31] (p. 12) state that SAT is "the extent, to which its users (i.e., teachers and students etc.) are contented with the system functionalities, for a productive learning experience, and how well is its performance, to meet the expectations of all the stakeholders." Literature has

confirmed the distinct association between e-learning system use and user SAT [10] and has acknowledged that if learners are more satisfied with an e-learning system, their intention to use these systems is substantially increased. Additionally, Cidral et al. [3] indicate that SAT is fundamental in evaluating the long-term adoption of e-learning systems. Moreover, the impact of SAT on ACP has been validated [32,66]. Specifically, a positive user experience with the e-learning system favorably affects learners' overall ACP.

Thus, Hypothesis 8 (H8) is proposed:

**Hypothesis 8 (H8).** *Students' SAT positively influences their ACP.*

*5.9. System Use and Academic Performance*

Salam and Farooq [31] (p. 13) state that use behavior demonstrates the degree to which "a user uses, the entire spectrum, of the available features, in a particular system, to fulfill his/her needs." More specifically, USE represents users' assessment of the overall use of a specific IS. In the literature, USE has been measured by, for example, the duration, nature, and frequency of use and users' perceived effectiveness and usefulness of IS to satisfy their requirements [67,68]. Net benefits represent all anticipated benefits of IS for an organization, a group, or an individual [69,70]. Contributions and outcomes of the e-learning system are weighted by the resultant impact on the organizational and individual levels.

This study focuses on individual benefits, and not organizational benefits. Hence, the construct of net benefits is replaced with ACP to add a more contextualized construction to the proposed model that fit the e-learning context and measured individual benefits for students. Maqableh et al. [27] define ACP as "students' ability to carry out academic tasks, and it measures their achievement across different academic subjects using objective measures such as final course grades and grading point average." Literature has reported an association between e-learning use and students' ACP [32] and has demonstrated that e-learning use supports students in their learning, for example, the provision of effective interaction, fast information interaction, and enhanced collaboration [29,35]. The actual use of the e-learning system designates that students acknowledge the use of the system as fulfilling their learning needs and helping them attain the learning outcomes. Accordingly, e-learning USE generates ACP.

Thus, Hypothesis 9 (H9) is proposed:

**Hypothesis 9 (H9).** *The use of the e-learning USE positively influences students' ACP.*

**6. Methodology**

*6.1. Participants and Target System*

This study used a sample of three private universities in Jordan that use Moodle, the target e-learning system, as a supplementary educational platform to traditional (face-to-face) teaching. Using both approaches—the online activities on Moodle and traditional teaching—in one course results in what is called a hybrid course, which is designed by instructors. The unit of analysis in this study was students who have used Moodle at least once in their learning. The accidental (convenience) sampling method was employed to select students. According to Gravetter and Forzano [71], this sampling method encourages participants to fill in the questionnaire based on their availability and willingness to participate. This study employed the accidental (convenience) sampling method because it is easier to use and faster in managing the survey than any other method.

Data were collected between December 2019 and March 2020, at the beginning of lockdown actions by the official authorities in Jordan. At that time, the actions of the official authorities in the country were viewed as precautionary measures rather than alarming ones. Nevertheless, the businesses around the country were severely affected and a considerable number of these businesses (including HEIs) were forced to move online. Such sudden and radical change occurred instantly. Hence, the survey of this study captures students' acceptance of e-learning systems in its actual condition without any specialized preparation and training.

Those students were selected based on convenience sampling. Data were collected using a web-based questionnaire survey. In principle, the survey was distributed at the end of a 3-month academic period (the length of the academic semester). The instructors were asked to share the survey link on the pages of their courses on Moodle. As a result, the survey's link was posted on 80 randomly selected courses on Moodle; in total, 1200 students were registered in these courses. Additionally, this study aimed to increase the response rate; thus, instructors were asked to send students a reminder 1 week after posting the survey. The survey was available for nearly 3 weeks. Of the 1200 students who received the survey, 577 students responded, resulting in a 48% response rate. After excluding incomplete and invalid responses (N = 40), 537 valid responses were obtained and subsequently used for validating and testing the research model. Table 1 presents the respondents' profile.

**Table 1.** Respondents' profile.

| Variable | | Frequency | Percentage |
|---|---|---|---|
| Gender | Male | 357 | 66% |
| | Female | 180 | 34% |
| Age in years | <20 | 202 | 38% |
| | 20–30 | 241 | 45% |
| | >30 | 94 | 17% |
| Enrolled course | Bachelor's | 493 | 92% |
| | Master's | 44 | 8% |
| Experience using the e-learning system | <1 year | 163 | 30% |
| | 1–2 years | 331 | 62% |
| | >2 years | 43 | 8% |

N = 537.

### 6.2. Instrument Design

As mentioned above, data were collected using a web-based questionnaire survey that comprised two main sections. The first section collected respondents' demographic data. The second section measured the nine proposed constructs in the research model. Particularly, the second section comprised 40 measurement items as each construct was measured by four items. All items were adopted from the related literature (Appendix A) and measured on a 5-point Likert scale from 1 (strongly agree) to 5 (strongly disagree).

Before final data collection, the validity and appropriateness of the questionnaire survey were evaluated using two procedures. In the first procedure, the measurement items were assessed by an expert academic panel of four individuals who have wide expertise in the field of IS. The assessment outcome reveals that the degree of agreement among the four members is 90.5%. Additionally, some suggestions to enhance the reliability and readability provided by the panel were considered. Second, a pilot study was performed on 60 students, which evaluated the reliability of the ten constructs. The results indicate that all constructs have adequate internal consistency as the Cronbach's alpha for each construct exceeded 0.7 [72].

### 7. Results

Structural equation modeling (SEM) was used to examine the relationships among the constructs of the proposed research model. SEM is deemed appropriate for data analysis because it enables scholars to analyze and manage complex models with many dependent and independent variables concurrently and comprehensively [73]. Accordingly, partial least square-SEM (PLS-SEM) was used for data analysis. Ketchen [74] stated that the PLS-SEM is adequate for validating predictive power and estimating significantly complex models. Accordingly, PLS-SEM has been used by many scholars in various fields such as e-commerce [75–81], information systems [82], e-government [83,84], and educational technology-related research [85–88]. Corresponding with Anderson and Gerbing [89],

data analysis using SmartPLS v.3.3.3 was performed throughout two key analytical stages. In the first stage, the reliability and validity of the measurement model were examined. Afterward, the structural model was tested to assess the estimation of path coefficients (hypothesis). The significance of the estimated path coefficients and loadings were assessed by the bootstrapping re-sampling procedure of 5000 re-samples. Before proceeding to the measurement model, the common method variance was evaluated. The test of Harman's one factor was performed to assess the presence of the CMV [72]. Accordingly, an exploratory factor analysis (EFA) was executed, and all measurement items were factorized into a one single factor. The result demonstrates that ten factors emerged, as none of these factors accounted for ≥50% of variance among the measurement items. The construct of academic performance (ACP) had the highest variance explained and accounted for 30.2% of the total variance. Hence, the absence of CMV was evident.

## 7.1. Measurement Model

In this stage, the validity and reliability of the research model's constructs, and their corresponding measurement items were assessed. Following Hair et al. [72], tests were performed, including the reliability of internal consistency, discriminant validity, and convergent validity. Confirming the presence of convergent validity requires the items loading on the intended theoretical constructs to be 0.708, and the value of average variance extracted to be ≥0.5. Regarding internal reliability, composite reliability and Cronbach's alpha estimates are required to be ≥0.7. Table 2 presents all items that acquired a loading greater than the recommended value of 0.708; each construct had an AVE value ≥0.5 and Cronbach's alpha and composite reliability values higher than 0.7. These results confirm the presence of convergent validity and internal reliability in the dataset.

Discriminant validity was evaluated by employing two criteria. First, the Fornell and Larcker [90] criterion was applied. The result suggests that the AVE square root of each construct should be greater than its correlation with any other construct in the model. Table 3 demonstrates that this condition was satisfied, confirming the presence of discriminant validity.

Second, the heterotrait–monotrait ratio approach was employed [91]. In Table 4, all values are ≤0.85; thus, the results of the Fornell–Larcker criterion and that discriminant validity is present are confirmed. In summary, all measures used in this study demonstrated satisfactory validity and reliability.

## 7.2. Goodness of Fit (GOF)

The measurement model was assessed for a satisfactory goodness-of-fit (GOF); the overall model fit was evaluated by evaluating five main indices (Table 5). According to Henseler et al. [92] and Benitez et al. [93], the actual values of all GOF indices are within and satisfy the recommended values/conditions, indicating that the proposed model fits the dataset.

## 7.3. Structural Model

Structural model evaluation was used to examine the paths between the research model's constructs in regard to the proposed hypothesis, $R^2$ "coefficient of determination or predictive power," and predictive relevance ($Q^2$) [72]. However, before evaluating the proposed structural relationships, it is vital to examine collinearity to confirm that the regression outcomes are unbiased. Consequently, as Hair et al. [72] advises, collinearity was evaluated using the values of variance inflation factor (VIF) of inner model (among constructs). When confirming that collinearity is absent, VIF estimates should be close to or less than 3. Table 6 indicates that all constructs acquired a VIF less than 3; thus, collinearity issues were absent.

**Table 2.** Construct reliability and validity.

| Construct | Item | Loading | $\alpha$ | CR | AVE |
|---|---|---|---|---|---|
| Academic Performance (ACP) | APC1 | 0.93 | 0.94 | 0.96 | 0.85 |
| | APC2 | 0.92 | | | |
| | APC3 | 0.91 | | | |
| | APC4 | 0.92 | | | |
| Course Content Quality (CCQ) | CCQ1 | 0.85 | 0.90 | 0.93 | 0.77 |
| | CCQ2 | 0.89 | | | |
| | CCQ3 | 0.88 | | | |
| | CCQ4 | 0.90 | | | |
| Educational System Quality (ESQ) | ESQ1 | 0.89 | 0.90 | 0.93 | 0.77 |
| | ESQ2 | 0.88 | | | |
| | ESQ3 | 0.87 | | | |
| | ESQ4 | 0.87 | | | |
| Instructor Quality (IQ) | IQ1 | 0.88 | 0.86 | 0.91 | 0.71 |
| | IQ2 | 0.82 | | | |
| | IQ3 | 0.85 | | | |
| | IQ4 | 0.83 | | | |
| Perceived Usefulness (PU) | PU1 | 0.94 | 0.94 | 0.96 | 0.85 |
| | PU2 | 0.93 | | | |
| | PU3 | 0.90 | | | |
| | PU4 | 0.93 | | | |
| Satisfaction (SAT) | SAT1 | 0.94 | 0.96 | 0.97 | 0.88 |
| | SAT2 | 0.94 | | | |
| | SAT3 | 0.93 | | | |
| | SAT4 | 0.94 | | | |
| Self-regulated learning (SRL) | SRL1 | 0.82 | 0.87 | 0.91 | 0.72 |
| | SRL2 | 0.88 | | | |
| | SRL3 | 0.83 | | | |
| | SRL4 | 0.85 | | | |
| Support Service Quality (SSQ) | SSQ1 | 0.84 | 0.86 | 0.91 | 0.71 |
| | SSQ2 | 0.82 | | | |
| | SSQ3 | 0.85 | | | |
| | SSQ4 | 0.86 | | | |
| Technical System Quality (TSQ) | TSQ1 | 0.86 | 0.88 | 0.92 | 0.74 |
| | TSQ2 | 0.87 | | | |
| | TSQ3 | 0.85 | | | |
| | TSQ4 | 0.86 | | | |
| System Use (USE) | USE1 | 0.89 | 0.90 | 0.93 | 0.78 |
| | USE2 | 0.86 | | | |
| | USE3 | 0.89 | | | |
| | USE4 | 0.89 | | | |

$\alpha$: Cronbach's alpha, CR: composite reliability, AVE: average variance explained.

$R^2$ and $Q^2$ were used to assess the predictive accuracy of the proposed model. As Hair et al. [94] suggests, all paths were tested by running a 5000 bootstrap re-samples procedure. Additionally, the blindfolding procedure was performed to calculate $Q^2$ estimates. Table 7 demonstrates that all the dependent variables possess a $Q^2$ estimate greater than 0, and the $R^2$ are all higher than 0.25, indicating that the proposed model possesses adequate predictive accuracy.

**Table 3.** Fornell and Larcker's test.

|  | IQ | APC | CCQ | ESQ | PU | SAT | SRL | SSQ | TSQ | USE |
|---|---|---|---|---|---|---|---|---|---|---|
| IQ | * **0.84** | | | | | | | | | |
| APC | ** 0.63 | **0.92** | | | | | | | | |
| CCQ | 0.59 | 0.63 | **0.88** | | | | | | | |
| ESQ | 0.62 | 0.65 | 0.60 | **0.88** | | | | | | |
| PU | 0.64 | 0.69 | 0.70 | 0.64 | **0.92** | | | | | |
| SAT | 0.65 | 0.66 | 0.66 | 0.67 | 0.69 | **0.94** | | | | |
| SRL | −0.56 | −0.58 | −0.59 | −0.63 | −0.62 | −0.62 | **0.85** | | | |
| SSQ | 0.55 | 0.56 | 0.60 | 0.61 | 0.62 | 0.63 | −0.56 | **0.84** | | |
| TSQ | 0.62 | 0.60 | 0.62 | 0.59 | 0.66 | 0.65 | −0.57 | 0.61 | **0.86** | |
| USE | 0.62 | 0.67 | 0.64 | 0.64 | 0.69 | 0.63 | −0.62 | 0.61 | 0.63 | **0.88** |

* Numbers on the leading diagonal are the square root of AVE for each construct, ** correlation among the constructs, CCQ: course content quality, TSQ: technical system quality, IQ: instructor quality, SSQ: support service quality, ESQ: educational systems quality, SRL: self-regulated learning, SAT: satisfaction, PU: perceived usefulness, USE: system use, APC: academic performance.

**Table 4.** Heterotrait–monotrait test.

|  | IQ | APC | CCQ | ESQ | PU | SAT | SRL | SSQ | TSQ | USE |
|---|---|---|---|---|---|---|---|---|---|---|
| IQ | - | | | | | | | | | |
| APC | 0.70 | - | | | | | | | | |
| CCQ | 0.66 | 0.68 | - | | | | | | | |
| ESQ | 0.70 | 0.70 | 0.67 | - | | | | | | |
| PU | 0.71 | 0.73 | 0.76 | 0.69 | - | | | | | |
| SAT | 0.71 | 0.69 | 0.71 | 0.72 | 0.72 | - | | | | |
| SRL | 0.64 | 0.65 | 0.66 | 0.71 | 0.68 | 0.68 | - | | | |
| SSQ | 0.64 | 0.62 | 0.68 | 0.69 | 0.68 | 0.69 | 0.64 | - | | |
| TSQ | 0.71 | 0.65 | 0.69 | 0.66 | 0.73 | 0.71 | 0.65 | 0.70 | - | |
| USE | 0.70 | 0.73 | 0.71 | 0.72 | 0.75 | 0.68 | 0.69 | 0.69 | 0.70 | - |

CCQ: course content quality, TSQ: technical system quality, IQ: instructor quality, SSQ: support service quality, ESQ: educational systems quality, SRL: self-regulated learning, SAT: satisfaction, PU: perceived usefulness, USE: system use, APC: academic performance.

**Table 5.** Model fit.

| Index | Acceptable Value/Condition | Actual Value |
|---|---|---|
| SRMR (standardized root mean square residual) | <0.08 | 0.045 |
| d_ULS (unweighted least squares) | "d_ULS < bootstrapped HI 95% of d _ULS and d_G < bootstrapped HI 95% of d_G" | 0.435 |
| d_G (geodesic discrepancies) | | 0.357 |
| NFI (normed fit index) | >0.9 | 0.912 |

**Table 6.** Collinearity test.

| | | Independent Variables | | | | | | | | |
|---|---|---|---|---|---|---|---|---|---|---|
| | | IQ | CCQ | ESQ | PU | SAT | SRL | SSQ | TSQ | USE |
| Dependent Variable | APC | - | - | - | 2.41 | 2.07 | - | - | - | 2.11 |
| | PU | 2.08 | 2.15 | 2.29 | - | - | 2.00 | 2.06 | 2.20 | - |
| | SAT | 2.16 | 2.38 | 2.32 | 2.78 | - | 2.04 | 2.09 | 2.29 | - |
| | USE | 2.16 | 2.38 | 2.32 | 2.78 | - | 2.04 | 2.09 | 2.29 | - |

**Table 7.** Predictive accuracy.

| Construct | $R^2$ | $Q^2$ |
|---|---|---|
| APC | 0.576 | 0.48 |
| PU | 0.64 | 0.54 |
| SAT | 0.641 | 0.56 |
| USE | 0.61 | 0.47 |

In Table 7, SAT, USE, and PU explain 57.6% of the variance in ACP ($R^2 = 0.576$). Although seven constructs contributed to explaining 64.1% and 61% of the variance in SAT ($R^2 = 0.641$) and USE ($R^2 = 0.61$), respectively, five constructs contributed to explaining 64% of the variance in PU ($R^2 = 0.64$). According to Chin [95], such explanation power is classified as moderate to substantial. Furthermore, the procedure of PLS predict was performed to evaluate the predictive power [72]. Each indicator of the endogenous factors had a lower prediction error for the research model than LM, considering RMSE except for a few items (PU1, SAT3, USE4) (see Table 8). Such results associated with the positive $Q^2$ values for all these indicators, demonstrating that research model possesses medium to high predictive power.

**Table 8.** PLS predict assessment.

| Item | $RMSE_{PLS}$ | $Q^2\_Predict_{PLS}$ | $RMSE_{LM}$ | $RMSE_{PLS} < RMSE_{LM}$ | Predictive Power |
|---|---|---|---|---|---|
| APC1 | 0.366 | 0.464 | 0.375 | Yes | |
| APC2 | 0.355 | 0.48 | 0.359 | Yes | High |
| APC3 | 0.358 | 0.475 | 0.365 | Yes | |
| APC4 | 0.37 | 0.446 | 0.375 | Yes | |
| PU1 | 0.344 | 0.526 | 0.337 | No | |
| PU2 | 0.337 | 0.543 | 0.344 | Yes | Medium |
| PU3 | 0.331 | 0.559 | 0.343 | Yes | |
| PU4 | 0.347 | 0.515 | 0.354 | Yes | |
| SAT1 | 0.325 | 0.552 | 0.327 | Yes | |
| SAT2 | 0.322 | 0.576 | 0.328 | Yes | Medium |
| SAT3 | 0.34 | 0.523 | 0.337 | No | |
| SAT4 | 0.341 | 0.524 | 0.343 | Yes | |
| USE1 | 0.372 | 0.443 | 0.379 | Yes | |
| USE2 | 0.37 | 0.454 | 0.372 | Yes | Medium |
| USE3 | 0.357 | 0.485 | 0.358 | Yes | |
| USE4 | 0.384 | 0.409 | 0.376 | No | |

Regarding the path analysis in in Figure 3 and Table 9, the results indicate that H1–H5 and H7–H9 are supported; thus, H6 is not supported. Specifically, SRL generates a negative influence on PU, SAT, and USE. PU is the strongest predictor of APC ($\beta = 0.299$, $p = 0.000$); CCQ is the main predictor of PP ($\beta = 0.289$, $p = 0.000$); ESQ has the strongest effect on SAT ($\beta = 0.169$, $p = 0.001$); PU is the strongest predictor of USE ($\beta = 0.230$, $p = 0.000$).

**Table 9.** Path analysis summary.

| Hypotheses | Path | β | SD | *T* Statistics | *p* Values | Result |
|---|---|---|---|---|---|---|
| H1a | IQ → SAT | 0.155 | 0.050 | 3.167 | 0.002 | Supported |
| H1b | IQ → PU | 0.159 | 0.047 | 3.430 | 0.001 | Supported |
| H1c | IQ → USE | 0.113 | 0.044 | 2.519 | 0.012 | Supported |
| H2a | CCQ → SAT | 0.153 | 0.053 | 2.897 | 0.004 | Supported |
| H2b | CCQ → PU | 0.289 | 0.052 | 5.577 | 0.000 | Supported |
| H2c | CCQ → USE | 0.118 | 0.057 | 2.103 | 0.035 | Supported |

**Table 9.** Cont.

| Hypotheses | Path | β | SD | *T* Statistics | *p* Values | Result |
|---|---|---|---|---|---|---|
| H3a | ESQ → SAT | 0.169 | 0.051 | 3.320 | 0.001 | Supported |
| H3b | ESQ → PU | 0.111 | 0.051 | 2.158 | 0.031 | Supported |
| H3c | ESQ → USE | 0.151 | 0.052 | 2.935 | 0.003 | Supported |
| H4a | SSQ → SAT | 0.118 | 0.049 | 2.413 | 0.014 | Supported |
| H4b | SSQ → PU | 0.110 | 0.044 | 2.483 | 0.013 | Supported |
| H4c | SSQ → USE | 0.101 | 0.043 | 2.340 | 0.019 | Supported |
| H5a | TSQ → SAT | 0.119 | 0.052 | 2.284 | 0.022 | Supported |
| H5b | TSQ → PU | 0.180 | 0.048 | 3.843 | 0.000 | Supported |
| H5c | TSQ → USE | 0.106 | 0.047 | 2.67 | 0.023 | Supported |
| H6a | SRL → SAT | −0.117 | 0.047 | 2.488 | 0.013 | Not Supported |
| H6b | SRL → PU | −0.125 | 0.049 | 2.554 | 0.011 | Not Supported |
| H6c | SRL → USE | −0.130 | 0.047 | 2.743 | 0.006 | Not Supported |
| H7a | PU → USE | 0.230 | 0.060 | 3.864 | 0.000 | Supported |
| H7b | PU → SAT | 0.148 | 0.063 | 2.347 | 0.019 | Supported |
| H7c | PU → APC | 0.299 | 0.059 | 5.036 | 0.000 | Supported |
| H8 | SAT → APC | 0.265 | 0.055 | 4.818 | 0.000 | Supported |
| H9 | USE → APC | 0.295 | 0.051 | 5.767 | 0.000 | Supported |

SD: Standard deviation.

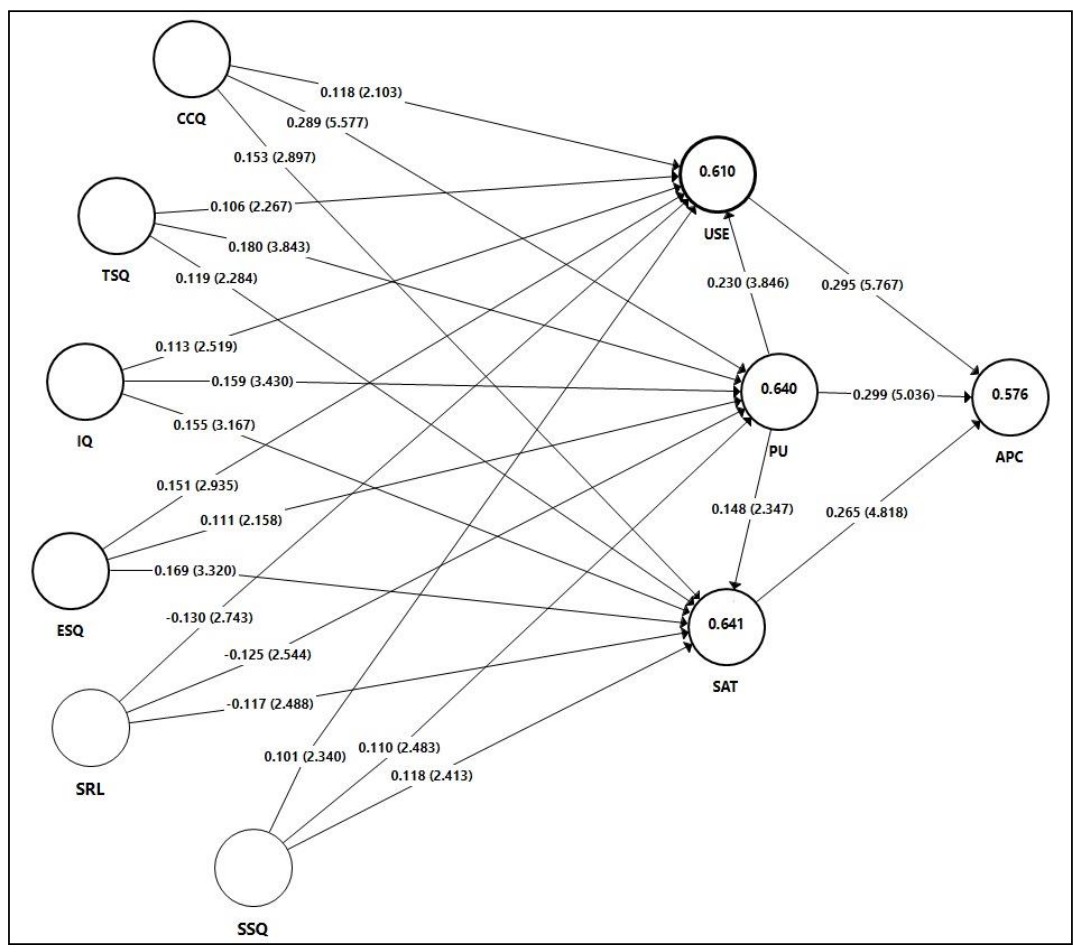

**Figure 3.** Structural model analysis. (*T* Statistics), CCQ: course content quality, TSQ: technical system quality, IQ: instructor quality, SSQ: support service quality, ESQ: educational systems quality, SRL: self-regulated learning, SAT: satisfaction, PU: perceived usefulness, USE: system use, APC: academic performance.

## 8. Discussion

The findings of this study confirm the positive effect of IQ on SAT and USE (similar to [62]) and on PU (similar to [10]). This indicates that providing timely, effective feedback from instructors using the e-learning system increases students' SAT and encourages them to use the e-learning system. Additionally, students are expected to develop high perceptions of e-learning usefulness when instructors respond to them in a timely manner and possess adequate technical skills and knowledge of course contents [45,61]. If instructors can manage all aspects of students' online learning and promptly respond to requests and questions using the e-learning system, students consider the system useful. As instructors are the main individuals recognized as a fundamental element by students in the e-learning environment, Kim et al. [96] indicate that instructors are viewed as a key success factor in e-learning environments because they increase students' SAT and motivate them to engage in learning opportunities. Instructors are considered content experts responsible for facilitating course delivery and managing students' learning [62]. Furthermore, instructors in e-learning plan the curriculum and adopt effective pedagogical strategies to take advantage of available technologies to ensure a successful implementation of e-learning [97]. When providing effective e-learning activities and responding to students' inquires and problems in a timely fashion, the use of e-learning by students is expected to be favored.

Consistent with the literature, CCQ is a significant enabler of students' SAT and USE [40,66] and an important facilitator of PU [10]. These findings suggest that when course content delivered by the e-learning system is considered well designed, regularly updated, and sufficiently comprehensive, students' PU and SAT increases. High-quality course content makes students feel contented with the e-learning system and thus encourages them to use the system. Additionally, when students believe that the system provides rich, frequently updated course content, and the degree of the courses' content can be customized to suit their individual needs, they consider the e-learning system to be useful for their learning. The availability of CCQ facets such as providing sufficient, clear, and accurate information; updated content; and content with an attractive, intuitive design is essential to have a pleasing experience with the e-learning system, and thus contributes to students' overall SAT. Similarly, logical, understandable organization of course components and content enables students to complete their learning tasks and responsibilities efficiently and more effectively and thus perceive the system as useful. Offering well-structured course designs helps students achieve the learning outcomes more effectively, enhancing SAT. Accordingly, the adequacy of courses' content encourages students' e-learning USE [37].

Similarly, the effect of ESQ on USE, SAT, and PU is significant. Although the significant influence of ESQ on USE and SAT has been confirmed in the literature [35,36], the significant influence of ESQ on PU and USE does not support the findings of Al-Fraihat et al. [10]. Our findings demonstrate that when the e-learning system provides high-quality functions to accomplish learning goals and tasks and facilitates the learning process, students perceive the usefulness of the system in delivering beneficial functions for effective learning. Moreover, students will perceive the system as useful if compatible functionalities are provided and students can continually access learning materials and contact their peers and instructors. These factors result in students' SAT with the system and increase their willingness to use the system.

The findings demonstrate that SSQ is a key determinant for PU, USE, and SAT. The significant influence of SSQ on SAT and USE is supported by [37], but the significant influence of SSQ on PU and USE contradicts the findings of Al-Fraihat et al. [10]. Our findings suggest that if students receive proper technical support services from a help desk or technical personnel, their SAT with the e-learning system increases. An appropriate quality of technical services provided by technical personnel to students significantly helps them develop a high usefulness perception toward the e-learning system and increases their use of it. In e-learning environments, students often encounter learning difficulties, for example, in conceptual understanding, technical issues, ease of access, and digital literacy [42]. If such difficulties are not resolved in a timely manner, the effectiveness of

and motivation for learning might decrease and have diverse effects on students' overall SAT, PU, and USE of the e-learning system. Markova et al. [98] highlight that student support may include technical or instructional support such that instructors assist students in overcoming issues encountered during online courses. This type of support, particularly when students encounter technical problems, is essential to overcome challenges that affect students' SAT [97]. Accordingly, the availability of helpful technical support leads to positive perceptions among students of SAT and e-learning system use. Additionally, technical support plays a fundamental role in revealing the novelties and key functionalities of e-learning systems to students, increasing students' PU of the system.

The findings demonstrate that TSQ has a positive impact on PU, US, and SAT. Notably, literature [3,40] has confirmed the positive influence of TSQ on SAT and USE [10] and that the relationship between TSQ and PU is nonsignificant. Our finding demonstrates that an e-learning system with high technical quality (i.e., easy to use, error free, user friendly) requires students to allocate minimal effort to learn and manage the system's functionalities; as a result, they can focus on the educational activities and content. Moreover, the appropriate technical design is a key determinant of how students can be effective in collaborating, interacting, and sharing learning material.

SRL is a key inhibitor of PU, SAT, and USE. This finding contradicts those in the literature [58,99]. This suggests that students are missing SRL, negatively influencing their SAT, usefulness perceptions, and e-learning USE. The e-learning environment is open in nature, and direct interaction with peers and instructors is low; thus, the responsibility of controlling learning processes shifts to the student [60]. As the e-learning system enables students to read and download learning materials at their convenience, students are obligated to control and manage their learning activities and processes. Hence, students with highly autonomous learning skills are more likely to be satisfied with the e-learning system than those with limited autonomous learning skills. Additionally, high SRL level allows students to recognize the value of the e-learning system's usefulness, such as the availability of learning material and ability to interact with peers and instructors at their convenience.

Consistent with the literature, the impact of PU on SAT [100], USE [101], and ACP [10] is significant. Such findings indicate that students are inclined to use the e-learning system if the system provides useful features that facilitate the completion of learning-related tasks efficiently and successfully. Moreover, students' SAT is increased by providing useful features that add value to the learning activities accomplished through the system. Thus, an increased perception of the e-learning system's usefulness will increase their ACP, and they will learn more effectively.

## 9. Research Implications

### 9.1. Theoretical Implications

E-learning adoption studies (based on TAM-related models) such as those on predicting e-learning adoption and postadoption behaviors (i.e., use, continued use) have ignored how such behaviors influence learning outcomes and ACP. Likewise, the D&M model has been criticized because of the limited theoretical support for relationships between determinants and behaviors of e-learning adoption. Accordingly, to overcome these limitations, this study suggests a novel e-learning success model that integrates the D&M model and TAM. Such integration considers the strengths of each model and thus overcomes the weaknesses in both models. The proposed model is comprehensive because it includes various perspectives related to quality, social factors, usefulness, acceptance, and benefits of using e-learning systems for the ACP of students. This study advances the literature and empirically examines the research model developed by combining a set of context-specific factors that are drivers of the success of e-learning systems. Particularly, five quality factors (CCQ, IQ, TSQ, ESQ, and SSQ) are suggested and empirically tested. These quality factors act as antecedents of PU, SAT, USE, and ACP. All these factors have demonstrated importance as valid measures that contribute to the recognition of e-learning success factors. Additionally,

unlike the literature, this study has introduced novel relationships not widely investigated in empirical studies. Particularly, new relationships are caused by the inclusion of SRL as a new context-specific factor. According to a review of the literature, this study is among the first studies to comprehensively investigate and empirically test the relationships between SRL and the aforementioned antecedents with PU, SAT, USE, and ACP in a single model. Finally, the proposed model demonstrates a considerable predictive power among PU, SAT, ACP, and USE. The proposed model moderately to substantially explains 57.6%, 64.1%, 61%, and 64% of the variance in ACP, SAT, e-learning use, and PU, respectively. Such predictive power is considerably higher than previous research (i.e., [1,37,66]).

*9.2. Practical Implications*

The findings demonstrate that IQ and support services affect the USE, PU, and SAT with the e-learning system. Consequently, instructors should receive adequate, constructive training before using the e-learning system [2]. This action will support instructors to acquire a broad understanding of the functionalities and features of the system and thus increase their confidence and efficiency when using the system. Additionally, instructors should master how to design online educational materials and how to teach online in manners that suit the nature and requirements of online learning environments. Providing technical and administrative supports to students to overcome learning difficulties or technical-related issues that may occur during their interaction with systems is also essential. Thus, a suggestion is that student support should be recognized as a key component of quality assurance that should be embedded in technology-enabled learning policies [42]. This may improve students' learning experience and overall perceptions and thus prevent poor interaction and frustration. Moreover, technical support might be delivered to students through training courses and manuals on how to use and manage the system in their learning. The quality of course content shows a positive influence on PU, SAT, and USE. Accordingly, designing the content of educational materials to fulfil e-learning settings is an essential aspect. As Pham et al. [41] indicate, educational materials should be relevant and continually updated to fulfil learning requirements and, more essentially, help students enjoy learning. Furthermore, it has been noted that incorporating virtual reality in e-learning platforms can significantly increase students' attractiveness, satisfaction, creativity, and motivation [102].

Educational and technical features of the e-learning systems are imperative aspects that impact PU, SAT, and USE. Hence, acknowledging that system features such as customization, integration between the system's components, and reliability and usability of the e-learning system should be enhanced to produce systems that are more appealing, reliable, intuitive, user friendly, customized, and convenient to navigate [10]. In addition to accessing the e-learning systems from ordinary websites, providing mobile access with appropriate designs would enhance students' SAT and sense of flexibility. Furthermore, the e-learning system must support students in developing collaborative, interactive learning environments through developing social networks with instructors and peers. Such collaboration and interactions may lead to higher learning effectiveness and, subsequently, higher grades. All these features contribute positively in enhancing students' PU, SAT, and USE.

Given the significant negative impact of SRL on students' PU, SAT, and the use of e-learning systems, universities should build modern pedagogical curricula that increase students' efficacy of self-regulation. This goal can be achieved by shifting from instructor to student-centered learning. Such an approach to learning requires students to possess self-regulatory competencies to be responsible for their learning. Student-centered learning in online learning environments should be supported with the use of pedagogical tools, such as multimedia tools and asynchronous and synchronous communication channels, to facilitate educational interactions with instructors and peers. Simultaneously, instructors are pivotal in enhancing students' SRL and so they must deliver the curricula and seek innovative methods to regularize and embed the skills of SRL in the learning activities [103,104]. Training courses might be useful in developing SRL strategies (i.e., time management,

planning, goal setting), especially at early stages in study programs [105]. Additionally, the developers of e-learning systems play an essential role in raising SRL competencies. They should focus on finding methods and solutions that support students in time and task management, personalized planning, and goal settings.

The findings reveal that students' PU of the e-learning system is a major determinant of SAT, USE, and ACP. What is critical is that e-learning systems deliver various valuable, beneficial tools and features that support students' learning. For instance, the ability to receive course-related announcements; download learning content; take online examinations; and access instant group discussion tools and channels that facilitate peer–instructor interaction, support, and feedback in real-time are strongly recognized as useful features [106]. Additionally, students' awareness of the benefits and usefulness of the e-learning system should be increased to enhance the popularity and use rates of the e-learning systems. Such awareness can be raised by conducting regular workshops and seminars to convey and demonstrate how the system benefits students.

Finally, SAT and USE are the main antecedents of ACP. Thus, students should support technology-enabled learning, and this support could be promoted by educating them on how e-learning USE can enhance their ACP. Moreover, as Abdous [107] argues, if the main driver for adopting e-learning is enhancing students' learning experience on-campus, institutional policies must be directed toward technology-enabled pedagogies and digital learning. Additionally, corresponding to Rajabalee and Santally [42], institutional leaders and e-learning policy makers should rely on learning analytics to improve their understanding of students' learning experiences and patterns.

## 10. Conclusions and Future Research

This study examines the key drivers of e-learning system success by developing a holistic research model that integrated TAM and the ISSM. The proposed model suggested a total of 23 hypotheses. To examine the research model, this study collected empirical data from students in three private universities in Jordan. The results indicated that 20 out of the 23 hypotheses were supported. Particularly, the suggested quality factors had a direct positive impact on the perceived usefulness, satisfaction, and use of e-learning. The quality factors included: instructor quality, course content quality, technical system quality, support service quality, and educational system quality. In addition, usage behavior, perceived usefulness, and satisfaction positively influenced students' academic performance. Surprisingly, self-regulated learning is a key inhibitor of e-learning systems success and negatively affects e-learning system's perceived usefulness, satisfaction, and use. Accordingly, e-learning stakeholders should introduce effective strategies to overcome the lack of students' self-regulated learning. In this study, the cross-sectional model captures users' perceptions and behaviors at a point in time. Further research is encouraged to employ longitudinal surveys considering the likelihood of a change in the preferences and perceptions of individuals as they gain experience. Further, the number of universities in the sample studied was limited. Such a limited sample raises questions on the generalization of the findings, especially to public universities. Accordingly, further research could extensively cover larger populations with different demographics and psychological and demographical attributes and a more representative sample. These steps would enhance the generalizability of the findings. This study, however, investigated three private universities and excluded public universities. In Jordan, more students attend public universities than private universities. Therefore, further research should consider public universities.

A suggestion is that cultures adopt different strategies in managing factors that affect e-learning adoption [35]. For example, although ease of use is perceived as essential in Eastern cultures, especially for developing countries, PU is more salient in Western cultures. Moreover, Eastern cultures are more socially oriented than Western cultures. This thus signifies the prominence of social factors in Eastern cultures. However, more attention should be paid to subjective norms, absorptive capacity, and mobility of individuals in Eastern cultures, not Western cultures. Consequently, further research should investigate

cultural effects. Finally, the proposed research model was tested using self-reported measures. Thus, future research should use objective measures to test the proposed model.

**Author Contributions:** Conceptualization, A.S.A.-A. and N.A.A.; methodology, A.S.A.-A.; software, A.S.A.-A.; validation, A.S.A.-A., O.H. and W.M.A.-R.; formal analysis, A.S.A.-A.; investigation, A.S.A.-A.; resources, A.A. and O.H.; data curation, A.S.A.-A.; writing—original draft preparation, A.S.A.-A.; writing—review and editing, O.H., W.M.A.-R., N.A.A. and A.A.; visualization, A.S.A.-A. supervision, A.S.A.-A.; project administration, A.S.A.-A.; funding acquisition, A.A. and A.S.A.-A. All authors have read and agreed to the published version of the manuscript.

**Funding:** The study was supported by a grant from the United Arab Emirates University (UAEU), Fund number: 31B121.

**Institutional Review Board Statement:** Not applicable.

**Informed Consent Statement:** Not applicable.

**Data Availability Statement:** Exclude.

**Acknowledgments:** We would like to gratefully acknowledge the assistance and support of the faculty members of the departments of Electronic Business and Commerce, and Management Information Systems at Al-Ahliyya Amman University for providing insightful feedback.

**Conflicts of Interest:** The authors declare no conflict of interest.

## Appendix A. Questionnaire Form

| Construct | Item | Source |
|---|---|---|
| Instructor Quality (IQ) | **IQ1**: "I use Moodle as recommended by my instructors." | [10] |
| | **IQ2**: "I think an instructor's enthusiasm about using Moodle stimulates my desire to learn." | |
| | **IQ3**: "I receive a prompt response to questions and concerns from my instructors in Moodle." | |
| | **IQ4**: "I think communicating and interacting with instructors are important and valuable in Moodle." | |
| Course Content Quality (CCQ) | **CCQ1**: "The content and information available in Moodle is timely." | [36] |
| | **CCQ2**: "The content and information available in Moodle is useful and easy to understand." | |
| | **CCQ3**: "The content and information available in Moodle can be relied upon." | |
| | **CCQ4**: "The content and information available in Moodle is accurate." | |
| Educational System Quality (ESQ) | **ESQ1**: "I believe that communication facilities have been effective learning components in my study." | [37] |
| | **ESQ2**: "Moodle provides evaluation components and assessment materials (e.g., quizzes, assignments)." | |
| | **ESQ3**: "Moodle provides me with different learning styles (e.g., flash animation, video, audio, text, simulation, etc.) and they are interesting and appropriate in my study." | |
| | **ESQ4**: "Moodle provides interactivity and communication facilities such as chat, forums, and announcements." | |
| Support Service Quality (SSQ) | **SSQ1**: "The IT services staff understands the specific needs of students." | [3] |
| | **SSQ2**: "I receive a satisfactory and timely response from the IT services staff." | |
| | **SSQ3**: "The IT services staff is available and cooperative when facing an error at Moodle." | |
| | **SSQ4**: "Moodle provides proper online assistance and help." | |

| Construct | Item | Source |
|---|---|---|
| Technical System Quality (TSQ) | **TSQ1**: "It is easy to understand the structure of Moodle and how to use it." | [37] |
| | **TSQ2**: "Moodle includes the necessary features and functions I need." | |
| | **TSQ3**: "Moodle navigation is easy to use." | |
| | **TSQ4**: "Overall, Moodle is easy to use." | |
| Self-regulated learning (SRL) | **SRL1**: "When it comes to learning and studying, I am a self-directed person." | [108] |
| | **SRL2**: "In my studies, I am self-disciplined and find it easy to set aside reading and homework time." | |
| | **SRL3**: "In my studies, I set goals and have a high degree of initiative." | |
| | **SRL4**: "I am able to manage my study time effectively and easily complete assignments on time." | |
| Perceived Usefulness (PU) | **PU1**: "Using Moodle enables me to accomplish my tasks more quickly." | [35] |
| | **PU2**: "Using Moodle improves my learning performance." | |
| | **PU3**: "Using Moodle helps me learn effectively." | |
| | **PU4**: "Overall Moodle is useful." | |
| Satisfaction (SAT) | **SAT1**: "I am satisfied with the performance of Moodle." | [1] |
| | **SAT2**: "I enjoy using Moodle in my study." | |
| | **SAT3**: "Moodle satisfies my educational needs." | |
| | **SAT4**: "Overall, I am pleased with the experience of using Moodle." | |
| System Use (USE) | **USE1**: "I use Moodle frequently." | [40] |
| | **USE2**:" I depend on Moodle in my study." | |
| | **USE3**: "I use Moodle regularly." | |
| | **USE4**: "On average, I spend a long time on using Moodle." | |
| Academic Performance (ACP) | **ACP1**: "Moodle has helped me to achieve the learning goals of the module." | [29] |
| | **ACP2**: "I had good grades in such courses where Moodle is used heavily." | |
| | **ACP3**: "Moodle makes communication easier with the instructor and other classmates." | |
| | **ACP4**: "Moodle is a very effective educational tool and has helped me to improve my learning process." | |

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
