# Peer review of "Developing a Holistic Success Model for Sustainable E-Learning: A Structural Equation Modeling Approach"

_sustainability, doi:10.3390/su13169453_

Round 1
Reviewer 1 Report
The contribution is purely processed. The authors drew from a relatively large number of relevant sources. I evaluate positively implementation and processing of research. The authors did not forget to pilot the research, the validity and reliability of the research tool are described.
Author Response
Thank you for your effort and time.
Reviewer 2 Report
There is a lot of work in this article . However the hypotheses, conclusions and research should be well and throroughly explained. What are the practical implications after these results and where should the future research continue this work.
The conclusions and the hypotheses should have a connection. Conclusions are important so I suggest you explain them thorougly to improve this article.
Author Response
Thank you for your constructive feedback. We would like to inform you that all the comments have been addressed.

Reviewer 3 Report
Dear authors,
First of all, I want to congratulate you on your research efforts. The paper is well-written and touches the essential parts for a paper to be considered for publication in this journal.
The e-learning field became again very attractive for scholars due to the new challenges brought by the pandemic context. There is plenty of acceptance and/or use of technology frameworks and theories that researchers adapt to the specific fields and develop new hybrid and/custom models. The current research entitled “Developing a Holistic Success Model for Sustainable E-learning: Structural Equation Modeling Approach” investigates empirically the major factors that contribute to the success of e-learning systems. The research PLS-SEM model combines the information system success and technology acceptance theories. The methodology and the size of the sample are adequate for this study. The literature is relevant and recent.
I want to point out several aspects, some being critical, since other ones are less important, but I suggest to be taken into considerations by the authors. They are mentioned below.
There is no evidence regarding the period of data collection. Please provide more details. If there is a research protocol behind it, please mention it. It is not a mandatory request. You may simply mention if the Helsinki Declaration’s ethical standards were accomplished, like in [1].
COVID-19 outbreak caused dramatic behavioral changes in every field. I noticed the authors cited several works related to these pandemic times. I suggest authors attack also possible side implications in their study. If the COVID-19 period covers the period of the study then these mentions should appear in several parts of the paper, including Abstract, Keywords, Introduction. Otherwise, please hit only the conclusive parts by studying and citing the already cited works from your manuscript and [1,2], [3,4].
The analysis is almost complete. You may also run the PLSpredict procedures, such as in [1,2], methodologically grounded in [5,6] or earlier works. In PLS-SEM this is more important than the model’s goodness of fit or blindfolding, even if prediction power is modest.
Please specify several reasons for using PLS-SEM in survey studies by pointing out works from different fields, such as information systems [7], e-commerce [8], but also e/m-learning-related fields [1,2] [9,10].
The absolute R-squared values are not a clear model performance indicator. You may compare yours with similar/worse/better ones from the literature. For instance, even if using a different TAM model (UTAUT2), the output for the system use in [1] is similar.
In TAM studies literature, the UTAUT2 should be mentioned [11].
In order to symbolize the paths, please use arrows between factors instead of the greater and equal sign.
The newest requirement regarding the minimum number of subsamples to be used in bootstrapping is 10,000 [5].
The citation style is not related to this journal. Please use [1], [2], …
The following citations were missed from the bibliography: Safsoufet al., 2020; Yumei & Qiongwei, 2017; Alzahrani et al. 2017; Zhang & Thompson, 2019; Ketchen, 2013.
There are several possible inconsistent citation pairs, namely (W. Lee, 2010) and (Lee, 2010); (Dorobaţ et al., 2019) and (Doroba et al., 2019); (Eom and Ashill 2018) and (Eom et al., 2012);
Anderson, J. C., & Gerbing, D. W. (1988) and Anderson and Gerbing (1998).
Please check “Affiliation 1, 2, 3…; “ – no need to specify the word “Affiliation”; “1.. Introduction”
Best regards and good luck with your research!
References
- Sitar-Taut, D.-A.; Mican, D. Mobile learning acceptance and use in higher education during social distancing circumstances: an expansion and customization of UTAUT2. Online Inf. Rev. 2021, ahead-of-print, doi:10.1108/OIR-01-2021-0017.
- Sitar‐Tăut, D. Mobile learning acceptance in social distancing during the COVID ‐19 outbreak: The mediation effect of hedonic motivation. Hum. Behav. Emerg. Technol. 2021, doi:10.1002/hbe2.261.
- Sun, Y.; Lin, S.-Y.; Chung, K.K.H. University Students’ Perceived Peer Support and Experienced Depressive Symptoms during the COVID-19 Pandemic: The Mediating Role of Emotional Well-Being. Int. J. Environ. Res. Public Health 2020, 17, 9308, doi:10.3390/ijerph17249308.
- Hou, J.; Yu, Q.; Lan, X. COVID-19 Infection Risk and Depressive Symptoms Among Young Adults During Quarantine: The Moderating Role of Grit and Social Support. Front. Psychol. 2021, 11, doi:10.3389/fpsyg.2020.577942.
- Hair, J.F.; Hult, G.T.M.; Ringle, C.M.; Sarstedt, M. A Primer on Partial Least Squares Structural Equation Modeling (PLS-SEM); 3rd ed.; Sage, 2022;
- Hair, J.F.; Risher, J.J.; Sarstedt, M.; Ringle, C.M. When to use and how to report the results of PLS-SEM. Eur. Bus. Rev. 2019, 31, 2–24, doi:10.1108/EBR-11-2018-0203.
- Hair, J.F.; Hollingsworth, C.L.; Randolph, A.B.; Chong, A.Y.L. An updated and expanded assessment of PLS-SEM in information systems research. Ind. Manag. Data Syst. 2017, 117, 442–458, doi:10.1108/IMDS-04-2016-0130.
- Mican, D.; Sitar-Tăut, D.-A.; Moisescu, O.-I. Perceived usefulness: A silver bullet to assure user data availability for online recommendation systems. Decis. Support Syst. 2020, 139, 113420, doi:10.1016/j.dss.2020.113420.
- Kumar, J.A.; Bervell, B.; Annamalai, N.; Osman, S. Behavioral Intention to Use Mobile Learning: Evaluating the Role of Self-Efficacy, Subjective Norm, and WhatsApp Use Habit. IEEE Access 2020, 8, 208058–208074, doi:10.1109/ACCESS.2020.3037925.
- Mehta, A.; Morris, N.P.; Swinnerton, B.; Homer, M. The Influence of Values on E-learning Adoption. Comput. Educ. 2019, 141, 103617, doi:10.1016/j.compedu.2019.103617.
- Venkatesh, V.; Thong, J.Y.L.; Xu, X. Consumer acceptance and use of information technology: Extending the unified theory of acceptance and use of technology. MIS Q. Manag. Inf. Syst. 2012, 36, 157–178, doi:10.2307/41410412.
Author Response
Thank you for your valuable feedback and quality comments. Most of the comments have been addressed.

Reviewer 4 Report
Reviewer’s Comments
(The reviewer use the file: sustainability-1319811-peer-review-v3.pdf)
Basically, I think that the aim and methodology of this paper are relevant to Sustainability. Especially this is an important theme regarding e-learning nowadays. However, there remain issues that need to be addressed.
- According to the content of the article, it is recommended that the author(s) can remove the subtitle and change it to another description. For example, the author(s) can join the discussion on the acceptance of technology and the success of information systems. Or join the relevant case study and so on. Because adding a methodology to the subtitle is not compelling. Suggest the author(s) can think about.
- The description of the abstract should be improved. For example, On page 1, lines 18-20."In higher education learning, e-learning systems have become renowned tools worldwide. The evident importance of e-learning in higher education has resulted in a prenominal increase in the number of e-learning systems delivering various forms of services." The author(s) should try to describe constructive and helpful comments that have improved the abstract, especially to respond to the conclusion. Maybe the author(s) can do it in a streamlined manner with an abstract.
- About Section Abstract, on page 1, line 21-24. " By relying on the related literature review, an extensive model is developed by integrating the information system (IS) success model and technology acceptance model (TAM) to illustrate key factors that influence the success of e-learning systems. " The keyword " information system (IS) success model " should be written as (ISSM).
- On page 2, lines 70-73. About " Al-Fraihat et al. [10] and Cidral et al. [3] have identified two prominent streams of e-learning adoption-related research. One stream focuses on adoption behaviors (e.g., system use [USE], adoption, intention to use, usability), technological aspects, and the system."
There are two prominent streams mentioned in this description. But the reviewer couldn't see the description of the other part.
And lines 73-77. "Nevertheless, because technology has become widely accessible and reliable, a more recent research stream has explored instructors’ and students’ interactions and attitudes as critical determinants of e-learning success. Accordingly, this study develops a holistic success model of e-learning that considers human and technological perspectives of e-learning." What does the author(s) mean by referring to a recent research stream in this paragraph? Whether it should be supplemented by adding relevant reference materials.
Based on the above question, the reviewer suggested that the author(s) can review and rewrite the content of the Introduction.
- On page 3, lines 91-93. About "Researchers of the second stream have been inclined to integrate various other frame works, such as the IS success model [25], theory of planned behavior [20], and TAM [17] along with the IS continuance model." Regarding "IS success model" and "theory of planned behavior", the abbreviated remarks can be used for the first time, and this abbreviation can be used for subsequent expressions. For example, "Researchers of the second stream have been inclined to integrate various other frame works, such as the IS success model(ISSM) [25], theory of planned behavior(TPB) [20], and TAM [17] along with the IS continuance model." It is suggested that the author(s) can re-examine all related nouns in the article. In this way, more concise words can appear in the article, and it can be more clear.
- On page 2, lines 79. About Section 2. Significance of the Study. Suggest the author(s) should change Section 2. Literature Review and Research Hypotheses. And added a subtitle 2.1 TAM or 2.1 Technology Acceptance Model on line 80. This will give the reader a clearer understanding of the key points in this section.
- Continuation of point 6, line 80 to line 120. The reviewer suggested that the content of this section should be rewritten and the relevant content that conforms to the title should be adjusted. If the title is TAM, keep the content of TAM. If the rest is related to ISSM, then go to the next section. It is also suggested that some content of this section can be adjusted to make relevant statements in the first section "Introduction", EX: lines 94 to 120.
- On page 3, lines 122. About Section 3. Information System Success Model. Suggest the author(s) should change Section 2.2 ISSM or 2.2 Information System Success Model.
- On page 4, lines 159. About Section 4. Theoretical Foundation. Suggest the author(s) should change Section 3. Research Model and Hypotheses. And added a subtitle 3.1 Research Model on line 160. Such an article structure and title can make the content clearer for readers.
And suggest that the author(s) can add a figure to present this research model. It would help the reader to know the architecture.
- On page 5, line 200. About Section 5. Hypothesis Development. Suggest the author(s) should change Section 3.2 Hypothesis Development. And on line 201, the subtitle 5.1. Instructor Quality. Suggest the author(s) should change Section 3.2.1 Instructor Quality. Subsequent, 5.2.-5.9. Followed to change to 3.2.2-3.2.9. Such an article structure and title can make the content clearer for readers.
- On page 5, line 233. About "Hc1: IQ positively influences the use of the e-learning system". It should be changed to "H1c: IQ positively influences the use of the e-learning system".
- On page 9, line 398. About Section 6. Methodology. Suggest the author(s) should change Section 4. Methodology. And on line 399, the subtitle 6.1. Participants and Target System. Suggest the author(s) should change Section 4.1. Participants and Target System. And on line 421, the subtitle 6.2. Instrument Design. Suggest the author(s) should change Section 4.2. Instrument Design.
- On page 10, line 439. About Section 7. Data Analysis. Suggest the author(s) should change Section 5. Result. And please add this paragraph to explain the common-method variance (CMV) of this research. It would be better let readers to understand whether the CMV's situation in this study is not serious.
- On page 10, line 456. About Section 7.1. Measurement Model. Suggest the author(s) should change Section 5.1. Measurement Model.
- On page 12, lines 478-481. About the analysis of heterotrait–monotrait ratio approach, Suggest the authors should try to add the reference. It would be better to present the situation of discriminant validity. The authors could reference the below literature.
Henseler, J., Ringle, C.M. & Sarstedt, M. A new criterion for assessing discriminant validity in variance-based structural equation modeling. J. of the Acad. Mark. Sci. 43, 115–135 (2015).
https://doi.org/10.1007/s11747-014-0403-8
Or the author(s) forgot to cite the relevant references in this paragraph, it is recommended to add the source of the relevant references.
- ON page 12, line 484. About Section 7.1. Model Fit Indices. Suggest the author(s) should change Section 5.2. Model Fit Indices. And Suggest trying to show the Goodness of Fit (GOF), one more indicator for readers to understand this study's Model Fit.
- On page 12, lines 494-501. Regarding the suggestions made by Hair et al. [72], the research framework must be a formative indicator. For example, The variance inflation factor (VIF) is often used to evaluate collinearity of the formative indicators. (Hair et al. [72], p10) According to this research framework, it should be a reflective indicator. If this research is not a formative indicator, it is recommended to delete the content from line 494 to line 499. And delete Table 6.
The author is needed to confirm that the research structure is a formative or reflective indicator, and then appropriately modify the relevant content.
- Regarding Table 3 on page 11 and Table 4 on page 12, it is recommended to add a note at the bottom of the table. List the full names of relevant abbreviations for readers to read and identify.
- On page 14, Figure 2. Structural model analysis. It is suggested that the relevant path only needs to show the β value, and does not need to show the T value (it can be indicated by *). For example, IQ------ 0.159*** ----->PU
And the R2 representation of related USE, PU, SAT, APC, please change to R2=0.XXX. For example, the R SQURE value of PU put R2=0.640 below the circle. The author(s) can put each construct in the circle.
And add a note at the end of the table: * p-value <0.05;** p-value <0.01; *** p-value <0.001.
This can increase readability and clarity.
The authors could reference the below literature. (See page9, Figure 2. PLS results of the research model.)
https://www.researchgate.net/profile/Chien-Liang-Lin/publication/351026007_Continuance_Intention_of_University_Students_and_Online_Learning_during_the_COVID-19_Pandemic_A_Modified_Expectation_Confirmation_Model_Perspective/links/60893bf28ea909241e2ca227/Continuance-Intention-of-University-Students-and-Online-Learning-during-the-COVID-19-Pandemic-A-Modified-Expectation-Confirmation-Model-Perspective.pdf
- On page 15, line 527. About Section 8. Discussion. Suggest the author(s) should change Section 6. Discussion and Conclusions. And added a subtitle 6.1. Discussion on line 528. Such an article structure and title can make the content clearer for readers.
- On page 17, line 619. About Section 9. Research Implications. Suggest the author(s) should delete the line. And on line 620, about change Section 9.1 Theoretical Implications. Suggest the author(s) should change Section 6.2. Theoretical Implications.
On page 17, line 647. About Section 9.2. Practical Implications. Suggest the author(s) should change Section 6.3. Practical Implications.
- On page 18, line 713. About Section 10. Conclusion and Future Research. Suggest the author(s) should change Section 6.4. Conclusion and Future Research.

Author Response
Reviewer 4
Comment1: “On page 5, line 233. About "Hc1: IQ positively influences the use of the e-learning system". It should be changed to "H1c: IQ positively influences the use of the e-learning system".
Response1: The comment has been addressed and corrected.
Comment2: “About Section Abstract, on page 1, line 21-24. " By relying on the related literature review, an extensive model is developed by integrating the information system (IS) success model and technology acceptance model (TAM) to illustrate key factors that influence the success of e-learning systems. " The keyword " information system (IS) success model " should be written as (ISSM)”.
Response2: The comment has been addressed and corrected.
Comment3: “On page 3, lines 91-93. About "Researchers of the second stream have been inclined to integrate various other frame works, such as the IS success model [25], theory of planned behavior [20], and TAM [17] along with the IS continuance model." Regarding "IS success model" and "theory of planned behavior", the abbreviated remarks can be used for the first time, and this abbreviation can be used for subsequent expressions. For example, "Researchers of the second stream have been inclined to integrate various other frame works, such as the IS success model (ISSM) [25], theory of planned behavior (TPB) [20], and TAM [17] along with the IS continuance model." It is suggested that the author(s) can re-examine all related nouns in the article. In this way, more concise words can appear in the article, and it can be more clear”.
Response3: The comment has been addressed and the suggested corrections have been made.
Comment4: “On page 10, line 439. About Section 7. Data Analysis. Suggest the author(s) should change Section 5. Result. And please add this paragraph to explain the common-method variance (CMV) of this research. It would be better let readers to understand whether the CMV's situation in this study is not serious”.
Response4: the section title has been changed from “data analysis” to “results”. Additionally, the common-method variance (CMV) test has been added. The following statement has been added.
“The test of Harman’s one factor was performed to assess the presence of the common-method variance (CMV) [72]. Accordingly, An exploratory factor analysis (EFA) was executed, and all measurement items were factorized into a one single factor. The result demonstrates that ten factors emerged, as none of these factors accounts for ≥50% of variance among the measurement items. hence, the absence of CMV was evident”.
Comment5: “On page 12, lines 478-481. About the analysis of heterotrait–monotrait ratio approach, Suggest the authors should try to add the reference. It would be better to present the situation of discriminant validity. The authors could reference the below literature. Henseler, J., Ringle, C.M. & Sarstedt, M. A new criterion for assessing discriminant validity in variance-based structural equation modeling. J. of the Acad. Mark. Sci. 43, 115–135 (2015). https://doi.org/10.1007/s11747-014-0403-8 Or the author(s) forgot to cite the relevant references in this paragraph, it is recommended to add the source of the relevant references”.
Response5: The comment has been addressed and the suggested reference has been added.
Comment6: “ON page 12, line 484. About Section 7.1. Model Fit Indices. Suggest the author(s) should change Section 5.2. Model Fit Indices. And Suggest trying to show the Goodness of Fit (GOF), one more indicator for readers to understand this study's Model Fit.”
Response6: the comment has been addressed and the suggested correction has been made.
Comment7: “On page 12, lines 494-501. Regarding the suggestions made by Hair et al. [72], the research framework must be a formative indicator. For example, The variance inflation factor (VIF) is often used to evaluate collinearity of the formative indicators. (Hair et al. [72], p10) According to this research framework, it should be a reflective indicator. If this research is not a formative indicator, it is recommended to delete the content from line 494 to line 499. And delete Table 6. The author is needed to confirm that the research structure is a formative or reflective indicator, and then appropriately modify the relevant content”.
Response7: We appreciate the reviewer’s comment, and we do confirm the validity of the comment that the collinearity evaluation must be perform for formative indicators. However, the collinearity evaluation using the VIF values is used for the inner model (among constructs), which in this case become (formative). We clarified this issue by stating that “collinearity was evaluated using the values of Variance Inflation Factor (VIF) of inner model (among constructs)”.
Comment8: “Regarding Table 3 on page 11 and Table 4 on page 12, it is recommended to add a note at the bottom of the table. List the full names of relevant abbreviations for readers to read and identify”.
Response8: the comment has been addressed and corrections have been made.
Comment9:” About Section Abstract, on page 1, line 21-24. " By relying on the related literature review, an extensive model is developed by integrating the information system (IS) success model and technology acceptance model (TAM) to illustrate key factors that influence the success of e-learning systems. " The keyword " information system (IS) success model " should be written as (ISSM)”.
Response9: the comment has been addressed and the correction has been made.
Comment10:” And suggest that the author(s) can add a figure to present this research model. It would help the reader to know the architecture.”.
Response10: the suggested figure has been added.
Comment11: “20. On page 15, line 527. About Section 8. Discussion. Suggest the author(s) should change Section 6. Discussion and Conclusions. And added a subtitle 6.1. Discussion on line 528. Such an article structure and title can make the content clearer for readers. 21. On page 17, line 619. About Section 9. Research Implications. Suggest the author(s) should delete the line. And on line 620, about change Section 9.1 Theoretical Implications. Suggest the author(s) should change Section 6.2. Theoretical Implications. On page 17, line 647. About Section 9.2. Practical Implications. Suggest the author(s) should change Section 6.3. Practical Implications. 22. On page 18, line 713. About Section 10. Conclusion and Future Research. Suggest the author(s) should change Section 6.4. Conclusion and Future Research”.
Response11: we appreciate all these comments. However, we believe that the current sequence that we followed give a clear flow of the contents of each title (or sub-title). Further, the current sequence is adopted from well-established research in a high-quality Journals.

Round 2
Reviewer 3 Report
Dear authors,
First of all, I want to congratulate you on your research efforts. The revise increased the quality of the paper.
There are several minor changes to be considered.
"Hc1:" to be replaced with "H1c:".
However, I did not notice in the authors’ response attacking the following issues from the previous round.
“There is no evidence regarding the period of data collection. Please provide more details. If there is a research protocol behind it, please mention it. It is not a mandatory request. You may simply mention if the Helsinki Declaration’s ethical standards were accomplished, like in [1].
COVID-19 outbreak caused dramatic behavioral changes in every field. I noticed the authors cited several works related to these pandemic times. I suggest authors attack also possible side implications in their study. If the COVID-19 period covers the period of the study then these mentions should appear in several parts of the paper, including Abstract, Keywords, Introduction. Otherwise, please hit only the conclusive parts by studying and citing the already cited works from your manuscript and [1,2], [3,4].
The analysis is almost complete. You may also run the PLSpredict procedures, such as in [1,2], methodologically grounded in [5,6] or earlier works. In PLS-SEM this is more important than the model’s goodness of fit or blindfolding, even if prediction power is modest.”
Best regards and good luck with your research!
References
- Sitar-Taut, D.-A.; Mican, D. Mobile learning acceptance and use in higher education during social distancing circumstances: an expansion and customization of UTAUT2. Online Inf. Rev. 2021, ahead-of-print, doi:10.1108/OIR-01-2021-0017.
- Sitar‐Tăut, D. Mobile learning acceptance in social distancing during the COVID ‐19 outbreak: The mediation effect of hedonic motivation. Hum. Behav. Emerg. Technol. 2021, doi:10.1002/hbe2.261.
- Sun, Y.; Lin, S.-Y.; Chung, K.K.H. University Students’ Perceived Peer Support and Experienced Depressive Symptoms during the COVID-19 Pandemic: The Mediating Role of Emotional Well-Being. Int. J. Environ. Res. Public Health 2020, 17, 9308, doi:10.3390/ijerph17249308.
- Hou, J.; Yu, Q.; Lan, X. COVID-19 Infection Risk and Depressive Symptoms Among Young Adults During Quarantine: The Moderating Role of Grit and Social Support. Front. Psychol. 2021, 11, doi:10.3389/fpsyg.2020.577942.
- Hair, J.F.; Hult, G.T.M.; Ringle, C.M.; Sarstedt, M. A Primer on Partial Least Squares Structural Equation Modeling (PLS-SEM); 3rd ed.; Sage, 2022;
- Hair, J.F.; Risher, J.J.; Sarstedt, M.; Ringle, C.M. When to use and how to report the results of PLS-SEM. Eur. Bus. Rev. 2019, 31, 2–24, doi:10.1108/EBR-11-2018-0203.
- Hair, J.F.; Hollingsworth, C.L.; Randolph, A.B.; Chong, A.Y.L. An updated and expanded assessment of PLS-SEM in information systems research. Ind. Manag. Data Syst. 2017, 117, 442–458, doi:10.1108/IMDS-04-2016-0130.
- Mican, D.; Sitar-Tăut, D.-A.; Moisescu, O.-I. Perceived usefulness: A silver bullet to assure user data availability for online recommendation systems. Decis. Support Syst. 2020, 139, 113420, doi:10.1016/j.dss.2020.113420.
- Kumar, J.A.; Bervell, B.; Annamalai, N.; Osman, S. Behavioral Intention to Use Mobile Learning: Evaluating the Role of Self-Efficacy, Subjective Norm, and WhatsApp Use Habit. IEEE Access 2020, 8, 208058–208074, doi:10.1109/ACCESS.2020.3037925.
- Mehta, A.; Morris, N.P.; Swinnerton, B.; Homer, M. The Influence of Values on E-learning Adoption. Comput. Educ. 2019, 141, 103617, doi:10.1016/j.compedu.2019.103617.
- Venkatesh, V.; Thong, J.Y.L.; Xu, X. Consumer acceptance and use of information technology: Extending the unified theory of acceptance and use of technology. MIS Q. Manag. Inf. Syst. 2012, 36, 157–178, doi:10.2307/41410412.
Author Response
Reviewer 3
We would like to the reviewer for the constructive and quality comments. Such comments increased the quality of our research paper.
Comment1: “There are several minor changes to be considered. "Hc1:" to be replaced with "H1c:"
Response1: the typo has been addressed and corrected.
Comment2: “There is no evidence regarding the period of data collection. Please provide more details. If there is a research protocol behind it, please mention it. It is not a mandatory request. You may simply mention if the Helsinki Declaration’s ethical standards were accomplished, like in [1].”
Response2: The comment has been addressed and the period of data collection has been added. This can be seen in “6.1. Participants and Target System”.
Comment3: “COVID-19 outbreak caused dramatic behavioral changes in every field. I noticed the authors cited several works related to these pandemic times. I suggest authors attack also possible side implications in their study. If the COVID-19 period covers the period of the study then these mentions should appear in several parts of the paper, including Abstract, Keywords, Introduction. Otherwise, please hit only the conclusive parts by studying and citing the already cited works from your manuscript and [1,2], [3,4].”
Response3: The comment has been addressed and the covid-19 have been mentioned in the abstract, introduction, and keywords.
Comment4: “The analysis is almost complete. You may also run the PLSpredict procedures, such as in [1,2], methodologically grounded in [5,6] or earlier works. In PLS-SEM this is more important than the model’s goodness of fit or blindfolding, even if prediction power is modest.”
Response4: The required analysis has been performed this can be seen in section 7.3.

Reviewer 4 Report
The reviewer believes that this article has been well revised. But there are still some minor adjustments that need to be made.
- On page 5, lines 205-207. In Figure 2. The research model, According to your research framework, the final variable name should be academic performance (ACP), not net benefits (NEB). It is recommended to modify and adjust the variable name. In this way, you can fully present your research model and be consistent with the context of the article.
- On page 5, lines 205-207. About Figure 2. The research model, The reviewer suggested that the pictures and text should be enlarged appropriately so that the readers can see them more clearly.
- On page 11, lines 471-476. The worthy praise is that the part about CMV has been added to the article, but the disadvantage is that the part with the largest explanatory power of the single factor is not presented in the content. It is recommended to add it to the paragraph. Let readers know the extent of its influence more clearly.
- On page 14, line 552. The Table 8. PLSpredict assessment. About " PLSpredict assessment " should be written as " PLS predict assessment "
Author Response
We would like to thank you for such quality comments and feedback. Our response are highlighted below.
- On page 5, lines 205-207. In Figure 2. The research model, According to your research framework, the final variable name should be academic performance (ACP), not net benefits (NEB). It is recommended to modify and adjust the variable name. In this way, you can fully present your research model and be consistent with the context of the article.
Response: The final variable name has been changed from net benefits (NEB) to academic performance (ACP).
- On page 5, lines 205-207. About Figure 2. The research model, The reviewer suggested that the pictures and text should be enlarged appropriately so that the readers can see them more clearly.
Response: The whole figure has been modified and its text has been enlarged.
- On page 11, lines 471-476. The worthy praise is that the part about CMV has been added to the article, but the disadvantage is that the part with the largest explanatory power of the single factor is not presented in the content. It is recommended to add it to the paragraph. Let readers know the extent of its influence more clearly.
Response: The largest explanatory power of the single factor is included.
- On page 14, line 552. The Table 8. PLSpredict assessment. About " PLSpredict assessment " should be written as " PLS predict assessment "
Response: The typo has been corrected.